# Utilization of Bioflocculants from Flaxseed Gum and Fenugreek Gum for the Removal of Arsenicals from Water

**DOI:** 10.3390/ma15238691

**Published:** 2022-12-06

**Authors:** Deysi J. Venegas-García, Lee D. Wilson

**Affiliations:** Department of Chemistry, University of Saskatchewan, 110 Science Place, Thorvaldson Building (Room 165), Saskatoon, SK S7N 5C9, Canada

**Keywords:** mucilage-based flocculants, flaxseed gum, fenugreek gum, xanthan gum, coagulation–flocculation, roxarsone, arsenate, Box–Behnken Design

## Abstract

Mucilage-based flocculants are an alternative to synthetic flocculants and their use in sustainable water treatment relates to their non-toxic and biodegradable nature. Mucilage extracted from flaxseed (FSG) and fenugreek seed (FGG) was evaluated as natural flocculants in a coagulation–flocculation (CF) process for arsenic removal, and were compared against a commercial xanthan gum (XG). Mucilage materials were characterized by spectroscopy (FT-IR, ^13^C NMR), point-of-zero charge (pH_pzc_) and thermogravimetric analysis (TGA). Box–Behnken design (BBD) with response surface methodology (RSM) was used to determine optimal conditions for arsenic removal for the CF process for three independent variables: coagulant dosage, flocculant dosage and settling time. Two anionic systems were tested: **S1**, roxarsone (organic arsenate 50 mg L^−1^) at pH 7 and **S2** inorganic arsenate (inorganic arsenate 50 mg L^−1^) at pH 7.5. Variable arsenic removal (RE, %) was achieved: 92.0 (**S1**-FSG), 92.3 (**S1**-FGG), 92.8 (**S1**-XG), 77.0 (**S2**-FSG), 69.6 (**S2**-FGG) and 70.6 (**S2**-XG) based on the BBD optimization. An in situ kinetic method was used to investigate arsenic removal, where the pseudo-first-order model accounts for the kinetic process. The FSG and FGG materials offer a sustainable alternative for the controlled removal of arsenic in water using a facile CF treatment process with good efficiency, as compared with a commercial xanthan gum.

## 1. Introduction

Conventional treatment of water generally involves coagulation, flocculation, filtration, and disinfection methods [1]. The coagulation–flocculation (CF) process has been employed as a simple and effective way to destabilize, agglomerate and remove suspended particles from water and wastewater due to the effectiveness and colloidal properties of CF systems [2]. Currently, CF processes are facilitated with the use of inorganic coagulants [3]. Conventional flocculants are particularly important because they are efficient at low dosages and may form diverse types of colloidal nanostructured systems, which yield favourable flocs that result in efficient phase separation of pollutants through the CF process [4]. Among synthetic polymers, polyacrylamide as a well-known example of a flocculant that biodegrades poorly, whereas some of its degradation by-products and acrylamide residues have known toxicity [5]. Over the decades, the accessibility of synthetic polymers derived from non-renewable carbon sources has shifted to more sustainable alternatives as researchers explore more readily available natural biopolymer-derived materials from renewable biomass resources [6]. Natural biopolymers are generally acquired from renewable biomass (algae, plants, microbial and animal sources) that are either comprised of carbohydrate, lipids, or proteins [7]. Bioflocculants have gained increasing attention for water treatment due to their biodegradability, non-toxic properties, and effective flocculation performance, which are sometimes comparable with synthetic flocculants [8]. Flaxseed (*Linum usitatissimum* L.) is an ancient crop cultivated globally for its fiber and oil content [9]. The mucilage extracted from the outermost layer of flaxseed hulls has great potential utility as a biopolymer flocculant [10]. The flaxseed mucilage contains two major fractions: (*i*) rhamnogalacturonan-I (acidic fraction) and (*ii*) arabinoxylans (neutral fraction) [11]. Although flax has been used as a phytoremediation tool for the remediation of different heavy metals, there are limited reports on the use of flaxseed mucilage for wastewater treatment [12]. *Trigonella foenum-graceum* (fenugreek) is primarily cultivated in Asia, Northern Africa and the Middle East. Fenugreek is widely used as a flavoring agent and in folk medicine, where its seeds contain 23–26% protein, 6–7% fat and 58% carbohydrates (25% of which is dietary fiber), saponins, flavonoids and a gum that can be extracted from the endosperm of fenugreek seed [13]. Fenugreek gum (FGG) is mainly polysaccharide in nature, comprised of a typical type of galactomannan having a linear chain of β 1,4-linked D-mannose as the backbone, where a single unit of D-galactose is joined by α 1,6-linkage, with a 1:1 ratio D-mannose: D-galactose [14]. Limited studies are available that employ FGG in wastewater treatment, where available research indicates that FGG may serve as a promising material for CF-based processes [15,16].

Arsenic is a highly toxic element to animals and plants and it is widely distributed in the environment [17]. Long-term exposure to arsenic contaminated drinking water, even at low levels of exposure, will contribute risk to human health [18], including skin cancer, stomach cancer, respiratory tract cancer, and extensive liver damage [19]. Roxarsone (4-hydroxy-3-nitrobenzene arsonic acid) is an organoarsenical that was widely used as an antimicrobial feed additive to prevent parasitic diseases of poultry and greater livestock production through greater poultry weight gain [20]. A large fraction of roxarsone does not undergo metabolism that can be transferred to soil and water, whereas the inorganic by-products are the species that possess greater toxicity [21].

The main objective of this study was to evaluate the CF removal properties of flaxseed gum (FSG) and fenugreek gum (FGG) as natural flocculants for removal of organic and inorganic arsenic species, as compared with a commercial xanthan gum (XG) bioflocculant. The pH conditions were chosen herein to favour the formation of the diprotic As species, based on the pK_a_ values of arsenate and roxarsone, where maximum arsenic removal has been reported [22,23]. The Box–Behnken design (BBD) was employed to investigate the role of coagulant dosage, flocculant dosage and settling time on the removal efficiency (RE; %) of organic and inorganic arsenic species. The BBD method also affords determination of desirable operating conditions for achieving the maximum arsenic removal. Moreover, the adequacy of the model and the reliability of statistical analysis with various experimental parameters were determined by comparing the experimental and predicted RE (%) response for arsenic species in aqueous media.

## 2. Materials and Methods

### 2.1. Materials

All chemicals were of analytical reagent (AR) grade. Ferric chloride hexahydrate (97%), sodium hydroxide (99%), hydrochloric acid (35%), xanthan gum (from *Xantomonas campestris*), potassium antimony (III) tartrate hydrate (99%), antimony molybdate tetrahydrate (99.9%) and spectroscopic grade potassium bromide (99%) were purchased from Sigma-Aldrich (Oakville, ON, CA, USA). Sodium hydrogen arsenate heptahydrate (98%) was obtained from Alfa Aesar (Tewksbury, MA, USA). Potassium phosphate dibasic (99%) and potassium phosphate monobasic (99%) were obtained from Fisher Scientific (New York, NY, USA). L-ascorbic acid (99%) was obtained from BDH Chemicals Canada (Mississauga, ON, CA, USA). Flaxseed gum was obtained from the College of Agriculture and BioResources at the University of Saskatchewan (Saskatoon, SK, Canada). Fenugreek gum was obtained from Emerald seed products Ltd. (Avonlea, SK, Canada). Roxarsone was acquired from Haohua Industry Co. Ltd. (Changzhou, China) and was purified via recrystallization [23]. All materials were used as received unless specified otherwise. All stock solutions were prepared using 18 MΩ cm Millipore water.

### 2.2. Fourier-Transform Infrared (FT-IR) Spectroscopy

The FTIR spectra of the flocculants were obtained using a Bio-RAD FTS-40 IR spectrophotometer (Bio-Rad Laboratories, Inc., Philadelphia, PA, USA). Dried powder samples were mixed with pure spectroscopic grade KBr in a 1:10 weight ratio with grinding in a small mortar and pestle. The Diffuse Reflectance Infrared Fourier Transform (DRIFT) spectra were obtained in reflectance mode at 295 K with a resolution of 4 cm^−1^ over a spectral range of 500–4000 cm^−1^. Multiple scans were recorded and corrected relative to a KBr background.

### 2.3. Thermogravimetric Analysis (TGA)

Thermal profiles of the FSG, FGG and XG were obtained using a TA Instruments Q50 TGA system (New Castle, DE, USA) with a heating rate of 5 °C min^−1^ up to 500 °C and nitrogen as the carrier gas. The results reported herein are shown as first derivative (DTG) plots of weight loss (%) with temperature (wt.%/°C) against temperature (°C).

### 2.4. pH at Point-of-Zero-Charge (pH_pzc_)

The point-of-zero-charge (pH_pzc_) of materials was determined according to Kong & Wilson [24]. A stock solution of NaCl (0.01 M) was prepared and 20 mL portions were added into five vials (8-dram). The solution pH conditions of the samples were adjusted between pH 1 and 8 using aqueous NaOH or HCl solutions. The flocculants (65 mg) were added to each solution, and allowed to equilibrate for 48 h, before the final pH was recorded. The pH_pzc_ was estimated plotting final pH against initial pH, and the point of intersection of the resulting null pH refers to the point zero charge, pH_pzc_.

### 2.5. Coagulation–Flocculation Process

The coagulation–flocculation parameters were based on the experimental design matrix obtained from the BBD and the RSM, using a program-controlled conventional jar test Phipps & Bird PB-900 apparatus (Richmond, VA, USA) with six 2 L jars and stirrers. Two types of CF systems were carried out independently: (**S1**) organic arsenic (roxarsone) and (**S2**) inorganic arsenic (arsenate). Both systems ensured an initial concentration of 50 mg L^−1^ arsenic. Approximately 1 L of simulated arsenic-containing sample was added to the jar tester vessel and the pH was adjusted using 0.1 M NaOH or 0.1 M HCl to pH 7 and pH 7.5, for **S1** and **S2**, respectively. An aliquot of the arsenic solution was sampled to measure initial arsenic concentration. The CF process was carried out by adapting a reported method [25]. A predetermined amount of Fe (III) salt (25, 37.5 or 50 mg L^−1^) was added to the solution, followed by rapid stirring for 3 min at 295 rpm. Thereafter, the stirring rate was reduced to 25 rpm for 20 min. During this period, the bioflocculant (1, 60.5 or 120 mg L^−1^) was added within the first 5 min. Then, the stirring was stopped and different settling times (10–90 min) were tested. For **S1**, after dilution of the sample, 5 mL was used for UV−vis spectral analysis by adding 5 mL of phosphate buffer. A calibration curve of roxarsone was obtained (λ = 244 nm) with standard solution buffered using a Varian Cary-6000 Scan UV-vis spectrophotometer (Palo Alto, CA, USA). For **S2**, after dilution, 3 mL of sample was used for UV−vis spectral analysis by adding 0.5 mL of molybdate reagent [26]. After addition of the reagent to the arsenic sample, a blue colored complex formed after 20 min, prior to collection of the UV−vis absorbance values. A calibration curve of arsenate was obtained using the molybdate colorimetric method (λ = 900 nm) using a SPECTRONIC 200 visible spectrophotometer (Waltham, MA, USA). Experiments were repeated in duplicate, where the average value was reported. The removal efficiency (RE; %) of arsenic and the adsorption capacity, *q_e_* (arsenic; mg·g^−1^) were calculated by Equations (1) and (2), respectively.
(1)$$RE\;\left(\%\right)=\frac{C_{o}-C_{e}}{C_{o}}\times 100$$
(2)$$q_{e}=\frac{\left(C_{o}-C_{e}\right)\times V}{m}$$

Here, *C_o_* and *C_e_* are the initial and equilibrium concentrations (mg L^−1^) of arsenic, *V* is the volume (L), and *m* is the total weight (g) of the system (coagulant + flocculant).

### 2.6. Box–Behnken Experimental Design

The BBD method was used to determine the effects of key experimental operating variables on the arsenic removal and to determine the combined effect of the variables that yield the maximum arsenic removal (cf. Equation (1)). BBD is a response surface methodology, which is a collection of mathematical and statistical techniques that are useful for the modelling and analysis of systems wherein a response of interest is influenced by several variables, where the goal is to optimize this response [27]. Preliminary experiments indicate that three important operating parameters in the CF process that affect arsenic removal are the coagulant (FeCl_3_) dose, flocculant (FSG, FGG or XG) dose and settling time. These variables were chosen as the independent variables and designated as A, B and C, respectively. Coagulant dose (A) was changed between 25 and 50 mg L^−1^, flocculant dose (B) was varied between 1 and 120 mg L^−1^ and settling time (C) varied from 10 to 90 min. In Table 1, the experimental design involved the three parameters (A, B and C), each at three levels, coded −1, 0, and +1 for low, middle, and high concentrations, respectively.

To account for variability of the independent variables on RE (%), 12 points with 3 central points and no orthogonal blocking were carried out according to the statistical matrices developed by the RSM. Experiments were repeated in a duplicate manner, and the average value was reported. In correlating the arsenic removal efficiency (*ŷ*) with other independent variables (A, B and C), a response surface function was utilized (cf. Equation (3)). Where *ŷ* serves as the predicted response surface function, *b*_0_ is the model constant, *b*_1_, *b*_2_ and *b*_3_ linear coefficients, whereas *b*_12_, *b*_13_, and *b*_23_ are the cross-product coefficients, and *b*_11_, *b*_22_, and *b*_33_ are the quadratic coefficients in Equation (3). The response function coefficients were determined by regression analysis of the experimental data and the Minitab 19 DOE regression program.
*ŷ* = *b*_0_ + *b*_1_A + *b*_2_B + *b*_3_C + *b*_12_AB + *b*_13_AC + *b*_23_BC +*b*_11_A^2^ + *b*_22_B^2^ + *b*_33_C^2^(3)

### 2.7. Kinetic Studies

The adsorption of arsenate species (inorganic and organic) on the surface of adsorbent (coagulant + flocculant) is a dynamic process, where kinetic experiments reflect the change in the adsorption rate and time with the environmental conditions.

Kinetic models of adsorption such as the pseudo-first order (PFO) and pseudo-second order (PSO) models were used to examine the arsenic adsorption data. The term “pseudo” can be taken to imply that a rate law for adsorption is being expressed in terms of an adsorbed amount q (i.e., occupied sites of adsorption) onto a solid phase, in contrast to the concentration of the adsorbing species in the solution phase [28].

The non-linear forms of pseudo-first order (PFO) and pseudo-second order (PSO) kinetic models are defined by Equations (4) and (5), respectively:(4)$$q_{t}=q_{e}\left(1-exp^{-k_{1}t}\right)$$
(5)$$q_{t}=\frac{q_{e}{}^{2\;}k_{2}t}{1+q_{e}k_{2}t}$$

*q_t_* (mg g^−1^) and *q_e_* (mg g^−1^) indicate that the adsorption capacity of Fe(III)-flocculant towards arsenic removal at time (*t*), and at dynamic equilibrium. *k*_1_ (min^−1^), and *k*_2_ (mg g^−1^ min^−1^) are the respective rate constants of PFO and PSO parameters.

Kinetic studies were performed in situ using a one-pot method described by Mohamed & Wilson [29]. Briefly, a 600 mL beaker containing 500 mL of 50 mg L^−1^ arsenic solution that was mixed by magnetic stirring. A filter paper (Whatman no. 40) was folded into a cone and attached to the beaker that was immersed in the solution at a depth of 2 cm whilst stirring at 25 rpm (cf. Figure 1). An initial 2.5 mL aliquot was sampled from within the filter cone interior and added to a 25 mL volumetric flask. Sampling within the filter cone began at time (*t*) = 0, when the CF system was added, which continued at 1 min intervals for 10 min, then for a further 10 min at 2 min intervals, and finally for 40 min at 5 min intervals. At this point, stirring was stopped (*t* = 60 min) and sampling continued for a further 40 min at 10 min intervals. Sample aliquots were prepared for UV-vis spectral analysis, as discussed above for each system. The adsorption capacity at different times was determined using Equation (2).

## 3. Results and Discussion

### 3.1. FT-IR Spectroscopy

FT-IR spectra of FSG, FGG and XG are presented in Figure 2A. The FTIR spectra of FSG presented a broad and strong band observed at 3312 cm^−1^ revealed the presence of vibrational bands of the hydroxyl groups (–OH) that collectively form hydrogen bonds [30]. The existence of a weak vibrational band at 2932 cm^−1^ was ascribed to the presence of alkyl bonds of C–H (symmetric and asymmetric) [31]. The C=O asymmetric stretching vibration at 1604 cm^−1^ was attributed to the carboxyl group indicating the presence of uronic acids or bound water in FSG. The carboxyl group acts as a binding site for ions, which has a great influence on gelling and rheological properties of the flocculant. The absorption at 1420 cm^−1^ was referred to C–O stretching vibration and revealed the existence of uronic acids [32]. The area between 800 and 1200 cm^−1^ highlights the fingerprint region for carbohydrates, which was used as a good indicator to assess the structural differences of various gums. Furthermore, the peak at 1145 cm^−1^ supports the existence of the glycosidic linkages C–O–C and C–OH [33]. The IR spectrum of FGG shows a broad band at 3343 cm^−1^ associated with stretching vibrations of primary –OH group and a band at 2928 cm^−1^ due to the C–H stretching vibrations [15]. Three bands at 1452 cm^−1^ and 1377 cm^−1^ and 983 cm^−1^ correspond to C–H bond bending and at 1034 cm^−1^ is due to vibration of C–O–C stretching, respectively. The peak at 1144 cm^−1^ and 873 cm^−1^ can be assigned to the C–O and C–C bond stretching mode [14]. The band centered at 1230 cm^−1^ is indicative of the stretching vibration of C–O, which supports the presence of polysaccharides, due to their unique IR absorption properties [33].

The spectrum for XG shows a broad signal at 3360 cm^−1^, which indicates –OH bond stretching, and a sharp band at 2929 cm^−1^ relates to absorption of symmetrical and asymmetrical stretching of CH_3_ or CH_2_ groups [34]. The adsorption band at 1740 cm^−1^ describes –C=O stretching of carboxylic acids and esters, while the band at 1610 cm^−1^ denotes –C=O axial deformation in enols, such as β-diketones [35]. The band at 1412 cm^−1^ is attributed to –COO– groups and the absorption band at 1031 cm^−1^ indicates acetal groups [36]. The results described above display similar spectral profiles that are typical for IR spectra of such polysaccharide-based gums.

### 3.2. Solids ^13^C-NMR Spectroscopy

Figure 2B shows the ^13^C NMR spectra of FSG, XG and FGG. The ^13^C NMR spectrum of FSG presents a signature at 174.5 ppm, which relates to a carboxylate anion (–COO–) from a carboxymethyl substituent typical of FSG polysaccharides. The signature at 100.0 ppm relates to the anomeric ^13^C atoms of the polysaccharide, while C-2 and C-5 NMR lines of the polysaccharide overlap at 70.1 ppm. The primary carbon (C-6) resonance of FSG appears at 63.0 ppm. FSG contains two main polysaccharides that are acidic peptic-like with a neutral arabinoxylan polysaccharide, where the neutral carbohydrate moiety may contain uronic acid, depending on the overall molecular weight of FSG [37]. The resonances at 17.0 ppm represent the C-6 of the rhamnose. The ^13^C NMR spectrum of FSG results concur with those from a previous study [30,38,39]. The spectrum of the FGG presented a characteristic signature at 99.4 ppm, which is assigned to C-1 of galactose and mannose units. The peaks at 70.2 ppm and 62.1 ppm represent the C-2 and C-5 and C-6 of the galactose and mannose units, respectively. In addition, the signal at 80.5 ppm relate to C-1 of the β-D-mannose reducing chain end of FGG. The results obtained herein concur with other reported NMR spectra. The strong signal at 173.4 ppm in the XG spectrum is due to C=O from proteins, fatty acids and a pyruvated exopolysaccharide of the gum structure. The spectral lines at 25.1 ppm and 102.1 ppm represent the methyl carbon at C-1 of the glucose of the xanthan gum and the anomeric carbon (C-2) of the pyruvic acid ketal group, respectively. The broad and strong signal at 73.3 ppm relates to C-2, C-3 and C-5 of the glucose polymer chain. The peak located at 21.0 ppm shows the ^13^C (–CH_3_) group of acetate from proteins, fatty acids and exopolysaccharides [40].

### 3.3. Thermogravimetric Analysis (TGA)

Thermogravimetric analysis (TGA) is a simple method for studying the decomposition pattern and the thermal stability of the biopolymers. The TGA results herein are shown (Figure 3a) as a DTG profile (first derivative of mass with temperature) and weight loss (%) versus temperature (Figure 3b), to characterize the trends in thermal stability for FSG, FGG and XG. The weight loss events for the thermal decomposition of FSG, FGG and XG occurs in three steps: firstly at 25 °C to 150 °C due to desorbed water and volatiles, secondly at 150–250 °C due to the loss of functional groups (–OH, –COOH) and thirdly above 250 °C due to breakdown of the biopolymer backbone.

According to the TGA results in Figure 3a, the onset of degradation occurs at 275 °C, 300 °C and 290 °C for FSG, FGG and XG, respectively, in agreement with other reported TGA results [14,41,42].

### 3.4. pH at Point-of-Zero-Charge (pH_pzc_)

The pH_pzc_ parameter indicates the pH where the net surface charge of a material is zero, and it provides insight on the role of electrostatic interactions of a material surface with charged species. The pH_pzc_ of the FSG, FGG and XG were estimated according to the method reported by Kong & Wilson [24]. Figure 4 shows that the initial pH values of the XG, FSG and FGG are approximately 4.8, 4.9 and 5.9, respectively in agreement with other reports [43,44]. The surface charge of the adsorbents is positive at a pH below the pH_pzc_, where adsorption of OH^−^ ions and other anionic species occur because the ζ-potential of the sorbent is positive overall. Meanwhile, it is negative when the solution pH is above pH_pzc_, where adsorption of cation species occur due to deprotonation of the sorbent surface [20].

### 3.5. Box−Behnken Experimental Design

#### 3.5.1. Main Effects of the Independent Variables on the Response Functions

The removal of arsenic (RE; %) from water was investigated by a CF process. Three key operating parameters (coagulant dose, flocculant dose and settling time) were investigated using the BBD method. Plots depicting the main effects for the arsenic removal efficiency for system **S1** and system **S2** are presented in the Appendix A for the coagulant and flocculant dosage and settling time (cf. Appendix A). The corresponding plots illustrate important factors that are set at three levels for the designed experiment, which enables an in-depth analysis of the effects of factors on the CF process. These plots provide a preliminary estimate on the effects of independent variables: coagulant dose, flocculant dose and settling time on the response (RE; %). The statistical significance of the response function generated was verified by the F-test and ANOVA results (cf. Appendix A). Factors having a *p*-value < 0.05 that indicate statistical significance. The effect of increasing coagulant dosage from 25 to 50 mg L^−1^ (cf. Appendix A) shows that the arsenic removal increase with increasing coagulant dosage, as expected from other reports [45,46,47]. The maximum arsenic removal (%) achieved was 88, 87, 88.5, 76.1, 67.5 and 66.7 for **S1** arsenic removal with FSG (a), FGG (b) and XG (c), and **S2** arsenic removal with FSG (d), FGG (e) and XG (f), considering FeCl_3_ as the main factor. These results corroborate that the formation of the As-Fe oxides complex as the rate determining step in the process [48,49]. The role of flocculant dosage from 1 to 120 mg L^−1^ (cf. Appendix A) has a similar trend on arsenic removal (%) for **S1** and **S2** systems. An increased flocculant dosage results in greater arsenic removal (%) up to a maximum RE (%) value. Thereafter, the arsenic removal (%) decreases at higher flocculant concentration (120 mg L^−1^). This is probably due to the higher gum concentrations causing a vertical accumulation of the gum in the water column, which prevents efficient settling [14,50]. The effect on settling time for **S1** and **S2** was set from 10 to 90 min (cf. Appendix A). Each of the flocculants (FSG, FGG and XG) had a similar effect on the arsenic removal for **S1**. An increase in the settling times from 10 to 90 min led to greater arsenic removal (%), where small changes in arsenic removal and negligible changes in sludge volume was observed. For **S2**, the settling times from 10 to 90 min showed that greater settling time increased the arsenic removal (%) to a maximum value, where the arsenic removal (%) decreased at 90 min and beyond. According to these results, settling was achieved within 40–50 min. The rapid sedimentation indicated that the process is controlled by charge neutralization and bridging effects [51]. Maximum removal occurs when no more surface sites are available to the arsenate complex. Moreover, the adsorption time is limited by the settling of the gum, which sinks more rapidly as it gets heavier with adsorbed As-Fe oxide particles [52].

In order to explore the interactions of various major operating factors on arsenic removal by the flocculation process, the Response Surface Methodology (RSM) was employed. Three-dimensional (3D) response surface plots and 2D contour plots (Appendix A) were obtained from the model-predicted response by changing two independent variables within the experimental conditions at the same time as the third variable was fixed at a constant optimal level. These plots are useful to assess the interactive relationship between the independent variables and the response variable.

#### 3.5.2. Box−Behnken Analysis

The coefficients of the response function for arsenic RE (%) are presented in Table 2. Predicted values of RE (%) for arsenic were determined by the response functions with the obtained coefficients. A comparison of experimental and predicted values for coagulant dosage, flocculant dosage and settling time parameters for arsenic removal (%) are presented in the Appendix A (Appendix A for **S1**, and Appendix A for **S2**). Predicted and experimental values of arsenic RE (%) for **S1** and the various bioflocculants are in good general agreement.

The experimental data points in Box–Behnken experimental design were employed independently for each system, **S1** and **S2**, and three different flocculants (FSG, FGG and XG). The coefficients of determination (R^2^) values for system **S1** were 0.97, 0.99 and 0.95 for FSG, FGG and XG, respectively.

It is evident from Appendix A that the predicted responses are in good agreement based on the % error, which is less than the experimentally obtained responses. The coefficients of determination (R^2^) values for **S2** were 0.94, 0.99 and 0.98, which show that the response models are highly reliable.

#### 3.5.3. Confirmation and Validation

Optimal conditions for maximum arsenic removal were determined by the response models for the experimental data. The corresponding predicted arsenic removal (RE; %) for **S1**-FSG, **S1**-FGG, **S1**-XG, **S2**-FSG, **S2**-FGG, and **S2**-XG at these optimal conditions were 92.0 ± 0.6%, 92.3 ± 0.1%, 92.8 ± 0.1%, 77.0 ± 0.1%, 69.6 ± 0.6% and 70.6 ± 0.9%. The optimized conditions indicate a range between 32 and 36 mg L^−1^ of FeCl_3_ was required to achieve an optimal arsenic removal for the six different flocculant systems, showing that the As/Fe ratio is the main effect that governs the arsenic removal. For **S1**, similar concentrations of FGG (52 mg L^−1^) and of XG (50 mg L^−1^), were required to achieve optimal arsenic removal; meanwhile, a slightly higher concentration (64 mg L^−1^) was required for FSG. For **S2** the concentrations for FSG (60 mg L^−1^) and FGG (62 mg L^−1^) were similar, where the concentration required to maximize the arsenic removal efficiency for XG was lower (37 mg L^−1^). The settling time for **S1** and **S2** requires at least 50 min to achieve the highest arsenic removal, presenting the lowest settling time for the system **S2**-FGG with 51 min versus the longer **S1**-FSG (81 min). A comparison of experimental and predicted values at optimized conditions for coagulant dosage, flocculant dosage and settling time for arsenic removal (%) are presented in Table 3. The predicted and experimental values of arsenic removal (%) are in good agreement.

The validity of the statistical methodology and the experimental results was confirmed by performing additional experiments in duplicate for different conditions within the experimental design for each individual gum system. The selected conditions for the coagulant and flocculant dosage, and settling time are listed in Appendix A (cf. Appendix A) for **S1** and **S2**, respectively, including the predicted and the experimental results for RE (%) of arsenic. The experimental results for arsenic removal for each system are in good agreement with the predicted arsenic removal values.

Table 4 shows different mucilage-based and chitosan systems that serve as natural bioflocculants, where the conditions for arsenic removal, such as flocculant nature and dosage, coagulant nature and dosage, initial arsenic concentration and pH are listed. The RE (%) for FSG, FGG and XG presented herein show greater effectiveness based on the initial concentration.

### 3.6. Flocculation Kinetics

To study the effect of Fe (III)-flocculant for **S1** and **S2**, the one-pot kinetic experiment was carried out. Table 5 lists the kinetic parameters, where high correlation with the PFO model provides insight on the dependence of one of the components of the CF process. Based on reports in the literature, the initial concentration of arsenate is the main factor that governs the CF process [58,59]. Based on R^2^ values, the arsenic uptake by Fe (III)-FSG, Fe (III)-FGG and Fe (III)-XG systems obey the PFO model because of higher R^2^ values over the PSO model. By comparing the correlation coefficient (R^2^), the adsorption for arsenic onto the systems Fe(III)-gum is well described by the PFO model. Similar kinetic results for arsenic removal by biopolymers were previously reported [60].

The kinetics profile of Fe (III)-flocculant system used for the removal of arsenic is shown in Figure 5, where the two kinetic models (PFO and PSO) were used to fit the adsorption profiles for arsenic removal versus time. The adsorption process for arsenic occurs in two stages: a fast-initial adsorption stage, where ~90% of equilibrium adsorption capacity was achieved, and the subsequent slow adsorption phase, where the adsorption capacity gradually reached its maximum value for theequilibrium capacity.

## 4. Conclusions

This study focused on the use of flaxseed gum and fenugreek gum as flocculants, which were compared against commercial xanthan gum, for the removal of organic and inorganic arsenic species in water using a coagulation–flocculation (CF) process. Structural and physicochemical characterization of the gum flocculants employed spectroscopic (FT-IR, ^13^C NMR) and TGA to determine their relative composition and functional properties to guide the selection of materials for the flocculation process. A Box–Behnken statistical experimental design was used to determine the effects of coagulant dosage, flocculant dosage and settling time on the arsenic removal efficiency. The flocculation ability of the gum materials was investigated using an arsenic concentration of 50 mg L^−1^ and a fixed initial pH of 7 and 7.5, for the organic and inorganic arsenic species (roxarsone, **S1**; arsenate, **S2**), respectively. The Box–Behnken statistical experimental design and Response Surface Methodology were effective in determining the optimum conditions for arsenic removal by a CF process for several bioflocculants (FSG, FGG and XG). Experimental results in Table 3 demonstrated that the optimum coagulant dosage for maximum arsenic removal with ferric chloride was between 32 and 36 mg L^−1^ for **S1** and **S2,** which further validates the regression equations employed herein. Optimum flocculant dosages for maximum arsenic removal were found as follows: 64 mg L^−1^ (**S1**-FSG), 60 mg L^−1^ (**S2**-FSG), 52 mg L^−1^ (**S1**-FGG), 62 mg L^−1^ (**S2**-FGG), 50 mg L^−1^ (**S1**-XG) and 37 mg L^−1^ (**S2**-XG). The optimal settling time was estimated between 51 and 81 min. The kinetic results were studied in situ by using a one-pot kinetic method, where the PFO model provided a better fit relative to the kinetic profiles over the PSO model. FSG and FGG materials offer a sustainable alternative for the controlled removal of arsenic in water that employs the CF process. CF-based water treatment is a cost effective, environmentally friendly, facile and efficient treatment technology, as compared with other arsenic removal techniques. Therefore, the arsenic removal efficiency plus the low cost of FSG and FGG highlight the utility of these systems as promising bioflocculants for implementation in continuous water treatment systems.

## Figures and Tables

**Figure 1 materials-15-08691-f001:**
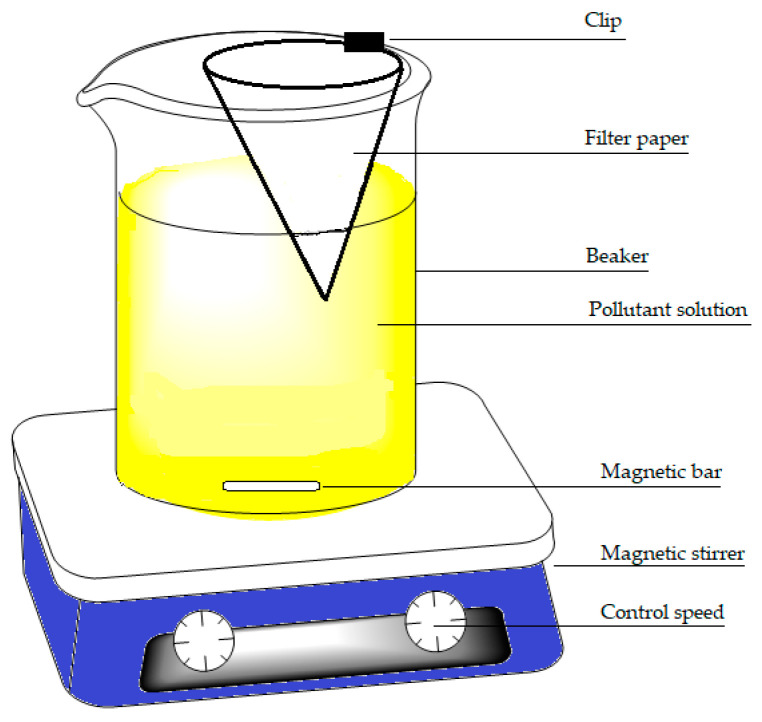
Illustration of the one-pot method for in situ kinetic studies.

**Figure 2 materials-15-08691-f002:**
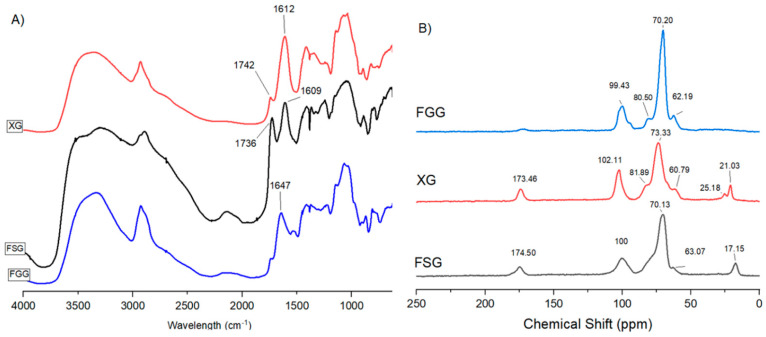
IR and ^13^C solids NMR spectra for flaxseed gum (FSG), fenugreek gum (FGG) and xanthan gum (XG): (**A**) FT-IR spectra, (**B**) ^13^C solids NMR spectra.

**Figure 3 materials-15-08691-f003:**
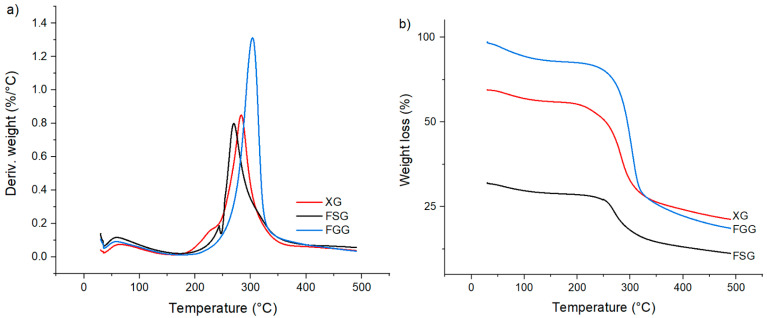
TGA results presented as a DTG profile (**a**) and weight loss versus temperature (**b**) for flaxseed gum (FSG), fenugreek gum (FGG) and xanthan gum (XG).

**Figure 4 materials-15-08691-f004:**
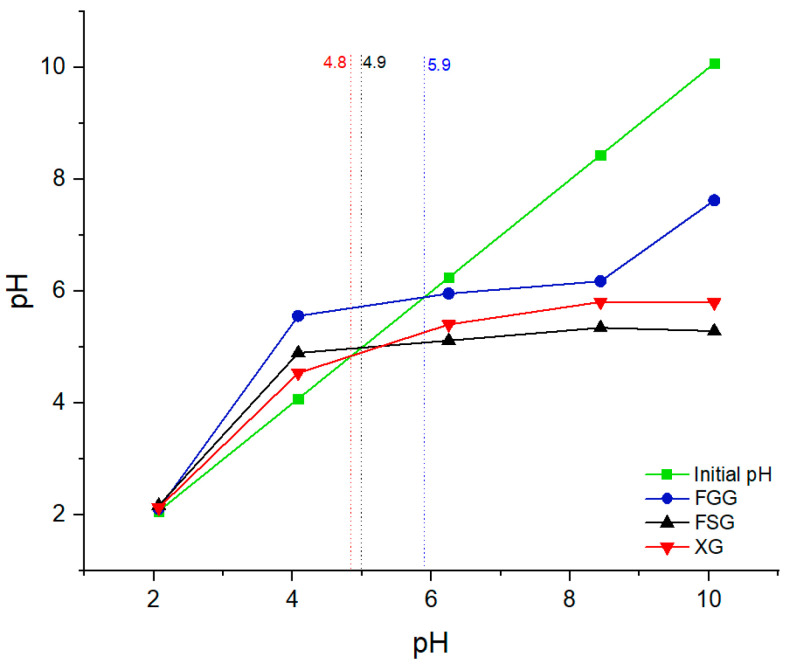
Zero-point-charge (pH_pzc_) as a function of variable initial pH for FSG, FGG and XG.

**Figure 5 materials-15-08691-f005:**
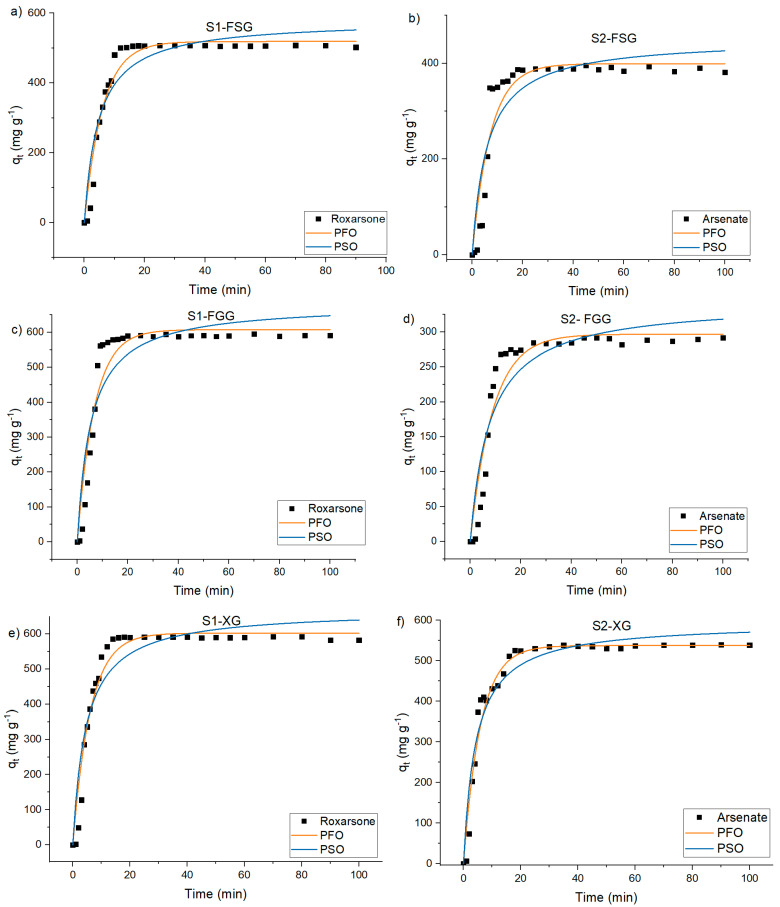
Kinetic profiles for the adsorption at 295 K: **S1**-FSG (**a**), **S2**-FSG (**b**), **S1**-FGG (**c**), **S2**-FGG (**d**), **S1**-XG (**e**), **S2**-XG (**f**), PFO and PSO kinetic-model best-fit results.

**Table 1 materials-15-08691-t001:** Levels of each factor for the Box–Behnken design method.

Independent Factors	Units	Symbol	Coded Levels		
			−1	0	1
FeCl_3_	mg L^−1^	A	25	37.5	50
Gum	mg L^−1^	B	1	60.5	120
Settling time	min	C	10	50	90

**Table 2 materials-15-08691-t002:** Coefficients of the response function for arsenic removal (RE; %) at 295 K.

Coefficients	S1			S2		
	FSG	FGG	XG	FSG	FGG	XG
Const	88.0	87.0	88.5	76.167	67.5	66.667
FeCl_3_ (A)	−10.25	−30.938	−18.25	−4.3125	−10.313	−9.1875
GUM (B)	2.125	−1.3125	−2.6875	−0.0625	3.25	−12.375
ST (C)	6.875	3.0	5.9375	6.0	1.8125	2.3125
AB	7.375	1.25	13.0	−7.0	4.625	4.0
AC	−4.375	−1.375	−1.0	−2.125	0.5	−0.125
BC	−1.375	1.125	3.375	−5.625	−0.625	2.75
AA	−28.938	−47.875	−34.563	−16.708	−12.625	−17.396
BB	−4.9375	−4.375	−18.188	−19.958	−27.5	−17.021
CC	−4.9375	−7.5	−6.4375	−15.583	−24.875	−6.8958

**S1** (organic arsenic), **S2** (inorganic arsenic), FSG (Flaxseed gum), FGG (Fenugreek gum), XG (Xanthan gum).

**Table 3 materials-15-08691-t003:** Results of confirmation experiments under optimized conditions at 295 K.

GUM	GUM Dose (mg L^−1^)	FeCl_3_ (mg L^−1^)	Settling Time (min)	As Removal (%) (50 mg L^−1^)	
				**S1** (Roxarsone)	
				Predicted	Experimental
FSG	64	35	81	91.8	92.0 ± 0.6
FGG	52	33	58	92.5	92.3 ± 0.1
XG	50	34	68	92.6	92.8 ± 0.1
				**S2** (Arsenate)	
				Predicted	Experimental
FSG	60	36	58	77.0	77.0 ± 0.1
FGG	62	32	51	69.4	69.6 ± 0.6
XG	37	34	54	70.6	70.6 ± 0.9

**Table 4 materials-15-08691-t004:** Comparison of maximum RE (%) of arsenic in aqueous medium using different biopolymers.

Flocculant	Concentration (mg L^−1^)	Coagulant	Concentration (mg L^−1^)	As (V) (mg L^−1^)	pH	RE (%)	Reference
*Aloe vera* gum	2	PAC	3	0.2–1	5	92.6	[53]
Chitosan	0.5	FeCl_3_		0.2–2	7	~100	[45]
*Opuntia ficus-* *indica gum*	350	-	-	0.002–0.01	5.9	70	[54]
-	-	FeCl_3_	27.029	1	5	98	[55]
-	-	Fe_2_(SO_4_)_3_	100	5	6	99	[56]
-	-	Al_2_(SO_4_)_3_	25–50	0.065–0.216	7–8	81	[57]
Flaxseed gum	64	FeCl_3_	35	50	7	90	This work
Fenugreek gum	52	FeCl_3_	33	50	7	90	This work
Xanthan gum	50	FeCl_3_	34	50	7	93	This work
Flaxseed gum	60	FeCl_3_	36	50	7.5	77	This work
Fenugreek gum	62	FeCl_3_	32	50	7.5	69.6	This work
Xanthan gum	37	FeCl_3_	34	50	7.5	70.6	This work

**Table 5 materials-15-08691-t005:** Adsorption kinetics parameters for **S1** and **S2** adsorption over FSG, FGG and XG at 295 K.

GUM	Pseudo-First-Order (PFO) Model			Pseudo-Second-Order (PSO) Model		
	*k_1_*	*q_e_*	R^2^	*k_2_*	*q_e_*	R^2^
	(min^−1^)	(mg g^−1^)		(min^−1^)	(mg g^−1^)	
	**S1** (Roxarsone)					
FSG	0.168	518.6 ± 10.8	0.952	3.6 × 10^−4^	580.3 ± 23.6	0.897
FGG	0.147	607.5 ± 16.7	0.924	2.7 × 10^−4^	682.3 ± 32.9	0.865
XG	0.163	602.6 ± 11.5	0.956	3.2 × 10^−4^	669.5 ± 25.1	0.901
	**S2** (Arsenate)					
FSG	0.138	398.9 ± 13.7	0.896	3.7 × 10^−4^	451.2 ± 25.6	0.842
FGG	0.112	296.7 ± 9.8	0.915	3.7 × 10^−4^	343.2 ± 19.6	0.869
XG	0.168	537.5 ± 8.6	0.967	3.9 × 10^−4^	595.1 ± 16.8	0.937

## Data Availability

The data presented in this study are available in this article and the accompanying Appendix A herein.

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
