# Peer review of "Utilization of Bioflocculants from Flaxseed Gum and Fenugreek Gum for the Removal of Arsenicals from Water"

_materials, 2022, doi:10.3390/ma15238691_

Round 1

Author Response

Authors Response to Reviewer Reports on MS ID:  materials-2002676

Please note that the Authors’ responses are in blue font. All corresponding changes were made to the manuscript in red font in the markup version.

Reviewer #1

The paper describes extensive experimentation using different techniques to predict maximal arsenic removal as a function of arsenic type, coagulant dosage, flocculant type, flocculant dosage, and settling time. Arsenic removal kinetics were found to be consistent with a “pseudo” first order model. I enjoyed reading the paper and have only minor suggested changes re: descriptions of the experimentation and how results are reported.

  1. The authors can easily provide error bars (95% CIs for example) on the reported removal efficacies (first provided on line 18). I understand if they do not want to give these in the abstract, but I do think they should be provided in the text (around line 293) and/or a table.

Response: Error was not listed in line 293, as these are calculated values from effect on main factors, but error was added to section “3.5.3 Confirmation and validation” and Tables SM9 and SM10.

  1. I thought that the authors did a nice job using the Supp. Materials to their advantage.
    However, I fell that Table S4, that gives estimates for the coefficients of the
    regression equations, should go into the text.

Response: The Table with coefficients was moved from Supp. Materials to the text.

  1. I think it is great that Tables 2 and 3 are provided. But I wonder whether Tables 2 and 3 ought to go to the Supp Materials. Table 1 already lists the Factor levels and implicitly defines the design points. The high R^2 values indicate that predictions are close to the data, so are Tables 2 and 3 really needed in main text? And removing them would help facilitate the previous point would be to move.

Response: Tables 2 and 3 were moved from the main text to Supp. Materials.

  1. I do not think that the general reader knows what the “one-pot” method is that is mentioned in the abstract, line 20. The authors do provide a description later in the paper. Perhaps add a few descriptive words in the abstract.

Response: The one-pot term was changed in the abstract by in-situ. Information from section “2.8 Kinetic studies” was move to section “3.6 Flocculation kinetics” in order to present the one pot kinetic and the results in a more organized form.

  1. The authors use “pseudo first order” in the abstract and text, and “pseudo second order” in text, please describe in text why “pseudo”.
    Please provide more details re: experimental design:
    Response: In section “2.8 Kinetic studies”, a brief explanation was included for the term “Pseudo”.

  1. Line 142, eq (1) generates enough data for the regression fit + pure error estimates only if the data Co and Ce are paired or if Co and Ce are means of blocks of replicates at each design point, please make this clear . If the latter, then how is the pure error calculated when the quadratic regression is fit to data generated from BBD?
    Response: Co refers to Initial (before adsorption) and Ce is the final (after adsorption) arsenic concentration in aqueous media. Using Eqn 1, the value for RE(%) can be calculated for each of the 15 experiments to proceed with the data analysis and the BBD optimization.

  1. Were the data collected over multiple experimental blocks? If so, were the blocks accounted for in the statistical analysis? If not, indicate that all 15 design points were conducted in a single experiment under identical conditions.
    Response: In section 2.7 that describes the Box-Behnken experimental design, a sentence was inserted to resolve the experimental runs: “To account for variability of the independent variables on RE (%), 12 points with 3 central points and no orthogonal blocking were carried out according to the statistical matrices developed by the RSM”.

  1. Line 146 when BBD is described in the methods, make clear there are 15 design points.

Response: In section 2.7, a sentenced describing the Box-Behnken experimental design was inserted to resolve the experimental runs: “To account for variability of the independent variables on RE (%), 12 points with 3 central points and no orthogonal blocking were carried out, according to the statistical matrices developed by the RSM”.

  1. In the methods, make clear how many data were collected at each BB design point and how many data were collected in total.
    Reporting BBD results:

Response: In section 2.7 Box-Behnken experimental design an additional statement was included: “Experiments were repeated in a duplicate manner”.

  1. Did the authors consider log-transforms of the their data? If so, then the ratio Ce/Co in equation (1) could be considered by a difference in logs.

Response: In the case of isotherm data that follow an exponential dependence (e.g., the Freundlich model), an argument for the logarithm dependence of adsorption on concentration (Ce) may be made. However, curve fitting analysis was carried out using nonlinear least squares fitting. Therefore, there is no statistical advantage to log transforms of the data in this study.

  1. When the optimal REs are reported at line 293, I wondered immediately what the values of coagulant dosage and flocculant dosage and settling time were that generated these optimal REs (via the model). I believe that these values are finally provided in Table 4. I suggest that the authors point out that the optimizing values of the independent are in Table 4 close to line 293 when the optimal REs are provided.

Response: Section “3.5.1. Main effects of the independent variables on the response functions” presents the effect of each factor over the RE (%). These values are calculated from Figure SM1, prior to optimization.

  1. What does Line 296 “considering FeCl3 as the main factor” mean? Is this a statement regarding what the most important factor is?

Response: In the literature, reports indicate that the coagulation process, destabilization of colloids, is the most important factor for the RE (%). The results presented herein provide support that the formation of the As-Fe oxides complex serve as the rate-determining step in the process.

  1. Table 2 “Experimental RE” gives values +/- values. Please define the values before and after the +/-, and how many replicates were used to generate them.

Response:  In Section 2.6, Coagulation-flocculation process, the calculation for the  RE(%) is described, where each experiment run was done by duplicate.

In addition, for section 2.7 Box-Behnken experimental design a sentence “Experiments were carried out in duplicate” was inserted.

  1. It is commendable that follow up experiments were performed to validate the regression equations (line 361). I suggest that this be mentioned in the abstract.

Response: A related comment was included in the conclusion section: “Experimental results in Table 3 demonstrated that the optimum coagulant dosage for maximum arsenic removal with ferric chloride was between 32 to 36 mg L-1 for S1 and S2, which further validate the regression equations employed.”

  1. Line 365-366 “The results for arsenic removal by experiment for each system are in good agreement with the predicted arsenic removal values“ is based on the reader visually comparing the predicted and experimental values in Tables S8 and S9. Can the authors provide a quantitative summary of this comparison, like an R^2 or how often confidence or tolerance intervals of the predictions capture the data?

Response: The % error calculation was added to Tables SM9 and SM10 (S8 and S9 before) with less than 5%.

Minor comments:

  1. Consider changing the notation on Line 162 and equation (3), to the standard $\hat Y$ to denote predictions. $Y$ is standard to denote actual data (“experimental RE”).

Response: Equation 3 and definition for Y was change to “ Å· ” based on the following reference: http://dx.doi.org/10.1080/00401706.1960.10489912  

  1. Perhaps change label Results at line 197 to Results and Discussion?

Response: The label was modified accordingly.

  1. In Supp Materials, indicate in Figure S1-S3 captions what the p-values in the panels indicate.

Response: Captions for Figures SM1-SM3 were modified to express the meaning of p value.

  1. Lines 315-321, I suggest that the authors point out the relevant plots in the Supp. Materials.

Response: Tables and plots presented in the Supporting Materials are mentioned in the text to point out their relevance.

In summary, the authors wish to acknowledge reviewer #1 for the constructive and insightful comments on this manuscript, along with the opportunity to improve the quality of this submission. We have further edited the manuscript for language, syntax, and clarity throughout to meet the high standards of this journal.

Reviewer 2 Report

Removing heavy metal ions from water is important for the wastewater treatment. Renewable sources are good candidates for the wastewater treatment attributing to their low cost, degradable and high specific surface area. The research of this manuscript is interesting and results are reliable. However, major revision is required and the comments are given below.

1.     Please pay attention to the writing of number and unit, a place should be added between the number and the unit. The units should be written in the same style. Please unify different styles, e.g. “mg/L” and “mg L-1”.

2.     Y axial should be added for TGA curves in Figure 2B.

3.     Please pay attention to the writing of subscripts and superscripts.

4.     Wastewater treatment is a hot topic and many absorbents with good performance are developed. It would be better to describe different absorbents in the introduction part for broad readers. More references are suggested to be cited, especially those newly published. Please refer and cite New Ulva lactuca Algae Based Chitosan Bio-composites for Bioremediation of Cd(II) Ions; Synthesis and Application of Granular Activated Carbon from Biomass Waste Materials for Water Treatment: A Review; MOFs meet wood: Reusable magnetic hydrophilic composites toward efficient water treatment with super-high dye adsorption capacity at high dye concentration.

5.     Please do some comparison on the arsenic removal performance with other methods or absorbents.

6.     Please double check the References. Page numbers are missing for some references.

Author Response

Authors Response to Reviewer Reports on MS ID:  materials-2002676

Please note that the Authors’ responses are in blue font. All corresponding changes were made to the manuscript in red font in the markup version.

Reviewer #2

Removing heavy metal ions from water is important for the wastewater treatment. Renewable sources are good candidates for the wastewater treatment attributing to their low cost, degradable and high specific surface area. The research of this manuscript is interesting and results are reliable. However, major revision is required and the comments are given below.

  1. Please pay attention to the writing of number and unit, a place should be added between the number and the unit. The units should be written in the same style. Please unify different styles, e.g. “mg/L” and “mg L-1”.

Response: The units were modified and are presented in a harmonized form.

  1. Y axial should be added for TGA curves in Figure 2B.

Response: Figure 2 was modified and the “Y” axis was added.

  1. Please pay attention to the writing of subscripts and superscripts.

Response: Subscripts and superscripts were edited throughout to address the reviewer query.

  1. Wastewater treatment is a hot topic and many absorbents with good performance are developed. It would be better to describe different absorbents in the introduction part for broad readers. More references are suggested to be cited, especially those newly published. Please refer and cite New Ulva lactuca Algae Based Chitosan Bio-composites for Bioremediation of Cd(II) Ions; Synthesis and Application of Granular Activated Carbon from Biomass Waste Materials for Water Treatment: A Review; MOFs meet wood: Reusable magnetic hydrophilic composites toward efficient water treatment with super-high dye adsorption capacity at high dye concentration.

Response:  We agree that citation of novel adsorbents is important. In the context of coagulant-flocculant systems, we have strived to cite the most relevant in relation to the current work (see Table 4).

  1. Please do some comparison on the arsenic removal performance with other methods or absorbents.

Response:  Table 5 compares arsenic removal with different biopolymers, some conditions such as initial arsenic concentration, coagulant nature and dosage, flocculant nature and dosage, pH and RE (%) are presented.   

  1. Please double check the References. Page numbers are missing for some references.

Response: The citations were checked to ensure that they meet the MDPI style requirements.

In summary, the authors wish to acknowledge reviewer #2 for the constructive and insightful comments on this manuscript, along with the opportunity to improve the quality of this submission. We have further edited the manuscript for language, syntax, and clarity throughout to meet the high standards of this journal.

Round 2

Reviewer 2 Report

The manuscript is well revised according to the suggestions and could be accepted.